# Is Colectomy Associated with the Risk of Type 2 Diabetes in Patients without Colorectal Cancer? A Population-Based Cohort Study

**DOI:** 10.3390/jcm10225313

**Published:** 2021-11-15

**Authors:** Chin-Chia Wu, Cheng-Hung Lee, Ta-Wen Hsu, Chia-Chou Yeh, Mei-Chen Lin, Chun-Ming Chang, Jui-Hsiu Tsai

**Affiliations:** 1Division of Colorectal Surgery, Dalin Tzu Chi Hospital, Buddhist Tzu Chi Medical Foundation, Chiayi 622, Taiwan; wccstillthought@gmail.com (C.-C.W.); daven88888@gmail.com (T.-W.H.); 2School of Post-Baccalaureate Chinese Medicine, Tzu Chi University, Hualien 970, Taiwan; kawa199190@gmail.com (C.-H.L.); yehcc0530@gmail.com (C.-C.Y.); 3Division of General Surgery, Dalin Tzu Chi Hospital, Buddhist Tzu Chi Medical Foundation, Chiayi 622, Taiwan; 4College of Medicine, Tzu Chi University, Hualien 970, Taiwan; 5Department of Chinese Medicine, Dalin Tzu Chi Hospital, Buddhist Tzu Chi Medical Foundation, Chiayi 622, Taiwan; 6Management Office for Health Data, China Medical University Hospital, Taichung 404, Taiwan; coolindm@gmail.com; 7College of Medicine, China Medical University, Taichung 404, Taiwan; 8Department of General Surgery, Hualien Tzu Chi Hospital, Buddhist Tzu Chi Medical Foundation, Hualien 970, Taiwan; 9Department of Psychiatry, Dalin Tzu Chi Hospital, Buddhist Tzu Chi Medical Foundation, Chiayi 622, Taiwan; 10Ph.D. Program in Environmental and Occupation Medicine, National Health Research Institutes and Kaohsiung Medical University, Kaohsiung 807, Taiwan

**Keywords:** colectomy, type 2 diabetes, cohort study

## Abstract

Type 2 diabetes might be influenced by colonic disease; however, the association between colonic resection and type 2 diabetes has rarely been discussed. This population-based cohort study explored the association between colectomy and type 2 diabetes in patients without colorectal cancer. A total of 642 patients who underwent colectomy for noncancerous diseases at any time between 2000 and 2012 in the National Health Insurance Research Database of Taiwan were enrolled. The enrolled patients were matched with 2568 patients without colectomy at a 1:4 ratio using a propensity score that covered age, sex, and comorbidities. The risk of type 2 diabetes was assessed using a Cox proportional hazards model. The mean (standard deviation) follow-up durations in colectomy cases and non-colectomy controls were 4.9 (4.0) and 5.6 (3.6) years, respectively; 65 (10.1%) colectomy cases and 342 (15.5%) non-colectomy controls developed type 2 diabetes. After adjustment, colectomy cases still exhibited a decreased risk of type 2 diabetes (adjusted HR = 0.80, 95% CI: 0.61–1.04). A stratified analysis for colectomy type indicated that patients who underwent right or transverse colectomy had a significantly lower risk of developing type 2 diabetes (adjusted HR = 0.57, 95% CI: 0.34–0.98). In the present study, colectomy tended to be at a reduced risk of type 2 diabetes in patients without colorectal cancer, and right or transverse colectomies were especially associated with a significantly reduced risk of type 2 diabetes.

## 1. Introduction

In Taiwan, the prevalence of diabetes is between 4.9 and 9.2% [1] and the age-adjusted prevalence of type 2 diabetes is 14.5% for men and 13.9% for women [2]. Diabetes mellitus is a serious chronic disease that requires an abundance of medical resources to treat [3]. Insulin resistance and pancreatic insufficiency are the mechanisms that lead to type 2 diabetes [4]. The complications of type 2 diabetes include nephropathy, neuropathy, and cardiovascular disease [5]. The endocrine pancreas is the main organ that regulates glucose through insulin and glucagon. Insulin and glucagon interact with the brain, liver, adipose tissue, muscle tissue, gastrointestinal tract and the gut–islet axis to maintain glucose homeostasis [6]. The gastrointestinal tract also participates in energy homeostasis through digestion, enteroendocrine functions, and the absorption of ingested nutrients in peripheral target organs [7].

Various hormones are released by the intestine after nutrient ingestion. The small intestine regulates glucose homeostasis mainly through glucagon-like peptide 1 (GLP-1), which is vital for the surgical control of type 2 diabetes [8]. GLP-1 is secreted by L cells and triggers insulin secretion from the pancreas. GLP-1 slows gastric emptying and promotes satiety. Glucagon secretion is inhibited by GLP-1, which depends on glucose [9]. The colon and rectum also contain L cells; the density of L cells increases from the right to the left of the colon and rectum [10], but the influence of colonic L cells on glucose homeostasis is still not well established.

The gut microbiota is believed to influence metabolism and the nervous system, which accounts for enteroendocrine function [11]. The colon is the main reservoir of the gut microbiota. The gut microbiota is important because of its association with many diseases, including cardiometabolic diseases [12]. The gut microbiota is associated with metabolic syndromes, such as obesity and type 2 diabetes [11,13], and cardiovascular disease [14].

In one study, colectomy in patients with colorectal cancer resulted in changes to the gut microbiota [15]. In patients with cancer and diabetes, colectomy resulted in the improvement of their diabetes [16]. In a Danish study, left colectomy was associated with a higher risk of type 2 diabetes [17], indicating that colectomy potentially influences the gut–islet axis. In a Taiwanese study of patients with colorectal cancer without chemotherapy, the gut microbiota and metabolic status were altered after curative colectomy [18]. In a Korean population-based study, cancer development increased the risk of subsequent diabetes [19]. In colonic resection for colorectal cancer, lymph node dissection offers a much more extensive eradication of metastasis than in resection for benign diseases [20]. For robust analysis on whether the colectomy truly influences the occurrence of type 2 diabetes, the study needs to remove the effect of cancer in itself.

This retrospective cohort study based on the Taiwan National Health Insurance Research Database (NHIRD) explored the association between colectomy and type 2 diabetes in patients without colorectal cancer.

## 2. Materials and Methods

### 2.1. Data Source

The Taiwan National Health Insurance (NHI) program is a single-payer insurance system managed by the Health Insurance Bureau. It covers approximately 99% of the population of Taiwan and has contracts with 97% of medical providers [21,22]. This study used NHIRD data. All diagnoses in the database were coded according to the International Classification of Diseases, Ninth Revision, Clinical Modification (ICD-9-CM).

We used the longitudinal database (LHID) 2000 derived from the NHIRD in this study. The National Health Research Institute of Taiwan randomly selected 1,000,000 NHI-insured patients in 2000 to establish the LHID; data on the patients were collected between 1995 and 2013. No statistically significant differences in age, sex, or health expenditure were found between the 1,000,000 individuals in the longitudinal database and all NHI-insured patients [23]. Detailed descriptions of how the LHID is sampled and how LHID data are collected have been previously published [24,25]. The study was approved by the Institutional Review Board (CMUH-104-REC2-115-R3) and informed consent was waived because of the use of previously stored de-identified medical information.

### 2.2. Study Population

We performed a retrospective cohort study of patients without colorectal cancer to explore the association between colectomy and type 2 diabetes. A flowchart of the study cohort collection is shown in Figure 1. Patients in the LHID 2000 who underwent colectomy (subtotal or total colectomy, right hemicolectomy or transverse colectomy, left hemicolectomy or sigmoid colectomy, proctectomy and partial colectomy) between 2000 and 2012 were selected by the National Health Insurance (NHI) Treatment codes (73045B, 73012B, 73011B, 73013B, 73014B, 73046B, 73048B, 74205B, 74206B, 74223B, 743213B, 74214B, 74216B, 74217B, 74222B, 73015B, 73047B, 73017B) for inclusion in the study. The index date was defined as the date of colectomy. The colectomy procedures and NHI treatment codes were illustrated in Figure 1 and Appendix A. We excluded the patients with colorectal cancer by the catastrophic illness certification. The catastrophic illness certifications are reviewed by experts according to pathological report and clinical staging, therefore the diagnosis is more accurate than ICD-9 cm [21]. The exclusion criteria were (1) colectomy treatment date out of the study period (2000–2012), (2) the presence of colorectal cancer, (3) patients with diabetes before the index date, (4) a diagnosis of diabetes mellitus before index date or within 3 months after index date, and (5) age <18 years or >100 years.

Non-colectomy patients were randomly selected from patients in the LHID 2000 who had no history of colorectal cancer and colectomy. We did not choose patients who underwent abdominal surgery without colectomy to form the control group because common abdominal procedures, such as cholecystectomy [26] and appendectomy [27], were reported to influence the gut microbiota. A propensity score was used to conduct frequency matching for colectomy cases with respect to non-colectomy patients in the control group at a 1:4 ratio according to age, sex, index year, and comorbidities. In the non-colectomy controls, the index date was the same as the matched case. The newly diagnosed type 1 attributed 0.56–0.66% of all diabetes patients in Taiwan [28]. Therefore, we defined type 2 diabetes (ICD-9-CM codes 250.x0 or 250.x2) was confirmed at two or more outpatient office visits or during one period of hospitalization within the study period. All participants were followed up until a diagnosis of type 2 diabetes after the index date, the end of follow-up in medical records, death, or the end of 2013.

The following comorbidities of all participants were evaluated: hypertension [29] (ICD-9-CM 401 or 405); hyperlipidemia [30] (ICD-9-CM 272); obesity [31,32] (ICD-9-CM m 278); chronic obstructive pulmonary disease [33] (ICD-9-CM 491, 493, or 496); chronic renal disease [34] (ICD-9-CM 582, 583, 585, 586, or 588), liver disease (except tumors) [35] (ICD-9-CM 571 or 572), anemia [36] (ICD-9-CM 280–285); and autoimmune diseases, including systemic lupus erythematosus and rheumatoid arthritis [37] (ICD-9-CM 710 or 714).

### 2.3. Statistical Analysis

Distributions of age, sex, and comorbidities are presented as numbers and percentages. Person years were calculated for each patient based on the time from index date to diagnosis of type 2 diabetes, death, or final follow-up (31 December 2013). Hazard ratios (HRs) and 95% confidence intervals (CIs) were estimated using the Cox proportional hazards model. The association between colectomy and type 2 diabetes was analyzed. The cumulative incidence of type 2 diabetes in the two cohorts was described using Kaplan–Meier plots and tested using the logarithmic rank test. All statistical analyses were performed using SAS statistical software (version 9.4; SAS Institute Inc., Cary, NC, USA). The Kaplan–Meier plot was plotted using R software. Statistical significance was determined using a two-tailed test (*p* < 0.05).

## 3. Results

After patients without colorectal cancer who did not meet the study criteria were excluded, 642 colectomy patients and 2568 non-colectomy patients were selected in this cohort study. In the colectomy patients, the leading five causes for colectomy were as follows: diverticular disease (21.8%), benign neoplasm of other parts of digestive system (15.1%), other disorders of intestine (11.8%), intestinal obstruction without mention of hernia (6.4%), and acute appendicitis (5.5%). The demographic characteristics and baseline comorbidities of the study cohorts are shown in Table 1. The mean age of the colectomy patients and non-colectomy patients at the time of presentation was 58.4 (standard deviation [SD], 18.1) and 58.3 (SD, 18.0) years, respectively. In total, 235 (36.6%) colectomy cases and 984 (38.3%) non-colectomy controls were women. The differences between both groups were nonsignificant with respect to age, sex, and other comorbidities.

In this 14-year cohort study, the mean (SD) follow-up times in the colectomy cases and the non-colectomy controls were 4.9 (4.0) and 5.6 (3.6) years, respectively; 65 (10.1%) colectomy patients and 398 (15.5%) non-colectomy patients had diagnosed type 2 diabetes during this follow-up period. Table 2 lists the risks of type 2 diabetes for colectomy in patients without colorectal cancer during this follow-up period. After adjusting for age, sex, and comorbidities, older colectomy patients were at a significantly higher risk of developing type 2 diabetes. In colectomy patients without colorectal cancer, patients aged ≥40 years had a significantly increased risk of developing type 2 diabetes than those aged <40 years (both *p* < 0.05). Being female (adjusted HR = 1.72, 95% CI: 1.08–2.86; *p* = 0.039) and having liver disease (adjusted HR = 1.79, 95% CI: 1.02–3.13; *p* = 0.043) were both associated with an independently increased risk of type 2 diabetes.

Figure 2 shows that the cumulative incidence of type 2 diabetes was significantly lower in the colectomy cohort than in the control cohort (Log-rank test, *p* = 0.024). As shown in Table 3, we stratified the study cohort by age, sex, and baseline comorbidities to explore the effect of colectomy on type 2 diabetes. The patients who underwent colectomy had a significantly lower risk of developing type 2 diabetes than controls who did not, with a HR of 0.74 (95% CI: 0.57−0.96; *p* = 0.024). After adjustment for age, sex, and comorbidities, colectomy cases still exhibited a lower risk for type 2 diabetes (adjusted HR = 0.80; 95% CI: 0.61–1.04). Male colectomy patients were associated with a reduced risk of developing type 2 diabetes (adjusted HR = 0.62, 95% CI: 0.43−0.89; *p* < 0.01). After adjustment, the colectomy cases were associated with a statistically significantly lower risk of developing hypertension, cerebrovascular disease, and heart disease. Furthermore, Table 4 indicates the association between risk of type 2 diabetes with different colectomy procedures in patients without colorectal cancer. After adjustment for age, sex, and comorbidities, the patients who underwent a right hemicolectomy or transverse colectomy had a significantly lower risk of developing type 2 diabetes, with a HR of 0.57 (95% CI: 0.34–0.98; *p* < 0.05). Other colectomy procedures were non-significantly associated with the risk of developing type 2 diabetes in this study.

## 4. Discussion

The results of this retrospective cohort study indicated that among patients undergoing non–colon cancer colectomy, those who underwent a right hemicolectomy or transverse colectomy were associated with a reduced risk of developing type 2 diabetes. However, subtotal or total colectomy, left hemicolectomy or sigmoid colectomy, and partial colectomy were not associated with the risk of developing type 2 diabetes.

We identified that an age of ≥40 years was a key risk factor for type 2 diabetes, which supports the results of a previous study [38]. Female patients had a higher risk of developing postoperative type 2 diabetes, which is consistent with a study of cardiac surgery patients [39]. Liver disease was demonstrated to produce endogenous insulin resistance and hyperinsulinemia [40], and we also found that liver disease was an independent risk factor for newly diagnosed postoperative type 2 diabetes. In the patients without colorectal cancer who underwent colectomy, the independent risk factors for type 2 diabetes were age, sex, and liver disease. This finding indicates that in older patients, female patients, and patients with liver disease, type 2 diabetes is a possible associated comorbidity after colectomy.

Several possible mechanisms can explain how colectomy alters glucose homeostasis, including endocrine secretion. Although the left side of the colon has a higher density of GLP-1-secreting cells than the right side of the colon, plasma GLP-1 levels remain unchanged after sigmoid colectomy or proctectomy in humans [41]. The results for the left hemicolectomy and sigmoid colectomy groups in our study are consistent with the finding that the risk of developing type 2 diabetes in patients undergoing these procedures does not change as a result of their colectomy. Gut dysbiosis is associated with an increased risk of metabolic disorders, such as glucose intolerance and obesity [42,43]. In gut dysbiosis, several universal bacteria that produce metabolically beneficial butyrate decrease while various opportunistic pathogens associated with type 2 diabetes increase [44]. Butyrate can stimulate GLP-1 secretion from L cells in the gut and regulate metabolism [45]. For example, the butyrate-producing bacteria Bifidobacterium [46] and Faecalibacterium prausnitzii [47,48] appear to decrease in patients who are overweight and have obesity or type 2 diabetes.

A key finding of this study was that the influence of colectomy on the risk of type 2 diabetes depends on the specific colectomy procedure. A reduction in type 2 diabetes risk after colectomy was statistically significant in the case of right hemicolectomy or transverse colectomy patients. This finding could be explained by the fact that the composition of microbiota differs by laterality [49] and leads to differences in nutrient fermentation. Carbohydrate fermentation occurs in the right colon, and protein fermentation occurs in the left colon [50]. Dominant fermentation of carbohydrates by the microbiota occurs in the right colon [50]. Removal of the main location of carbohydrate fermentation may explain why a lower risk of type 2 diabetes was associated with right hemicolectomy or transverse colectomy.

In a Taiwanese study on colorectal cancer patients without chemotherapy, the gut microbiota and metabolic status were altered after curative colectomy. The right hemicolectomy group had a higher occurrence of metabolic syndrome compared with the control group during long-term follow-up [18]. Our study excluded patients with colorectal cancer because they are at an increased risk of impaired glucose metabolism and pancreatic β-cell dysfunction [51]. Furthermore, patients with cancer tend to change their lifestyles and behaviors after cancer diagnosis [52]. Therefore, our study excluded patients with cancer to eliminate bias.

Contrary to Jensen’s study [17], our study demonstrated that the risk of type 2 diabetes decreased in the colon resection group. This study and this study highlighted how colon resection may be associated with the modulation of glucose homeostasis. A key difference is that our study population was from Southeast Asia. Metabolites interact with the environment of the colon to influence glucose homeostasis [53]. Dietary habits [54], the environment [55], and ethnicity may lead to different interactions between the gut microbiota [56] and its host. Another possible explanation is that diverticular diseases are associated with dysbiosis [57] in the gut and alterations in gut-associated endocrine regulation [58]. In previous studies, diverticular disease was associated with an increased risk of diabetes mellitus, including type 2 diabetes mellitus [59,60]. Diverticula are predominantly located on the right side of the colon in Taiwanese patients [61]. An explanation for why right-sided colectomy and transverse colectomy decrease the risk of type 2 diabetes is that dysbiosis is alleviated simultaneously with right-sided diverticulosis.

The strengths of our study were the use of a national database and a large sample size. We analyzed the risk of type 2 diabetes in patients without colorectal cancer after colectomy. The results are relevant to understanding the decreased risk of type 2 diabetes in such patients, especially in ethnic Chinese patients. To increase the accuracy of the diagnosis of type 2 diabetes, outpatient and inpatient care expenditure databases (containing diagnoses using ICD-9-CM codes) were used to confirm the diagnosis. Additionally, covariates, including those underlying common physical disorders, were considered. Despite our study’s retrospective nature, we used a case–control cohort to analyze the national database and avoided two major potential biases: selection bias and recall bias. However, our study had several limitations. First, risk factors for type 2 diabetes, such as smoking [62], body mass index [63,64], diet, and inactivity [65], are not included in the NHIRD. This is because body mass index is greatly associated with developing diabetes mellitus [63,64]. Although we were unable to obtain such information, we adjusted for comorbidities that are known risk factors for type 2 diabetes, including hyperlipidemia and hypertension. After adjustment, proximal colectomy remained a significant protective factor for type 2 diabetes. Secondary, the patients who underwent both bariatric surgery and colectomy were not managed in this study. Then, the patients were distributed evenly in the colectomy and non-colectomy groups. However, colectomy was associated with a decreased risk of newly diagnosed type 2 diabetes. Moreover, we were unable to determine what precise changes occurred after colectomy, either in the microbiota or the enteroendocrine system. This study was designed in the colectomy patients without cancer, and we could not compare it with the regular healthy Taiwanese population. Finally, dietary data were not available in the NHIRD. In Taiwan, although the main source of carbohydrates for most people is rice, we were unable to stratify patients by whether they were vegetarian. In this study, we only focused on comparison with the incidence of type 2 diabetes between colectomy and non-colectomy cases without colorectal cancer. Therefore, our study did not compare the risk of type 2 diabetes between colectomy cases with and without colorectal cancer. Although our study had several limitations, our finding that the type of colectomy procedure is associated with the risk of type 2 diabetes warrants further study and exploration.

## 5. Conclusions

In conclusion, we found that colectomy tended to be at a reduced risk of type 2 diabetes in patients without colorectal cancer; in particular, right or transverse colectomies were associated with significantly reduced risk of type 2 diabetes.

## Figures and Tables

**Figure 1 jcm-10-05313-f001:**
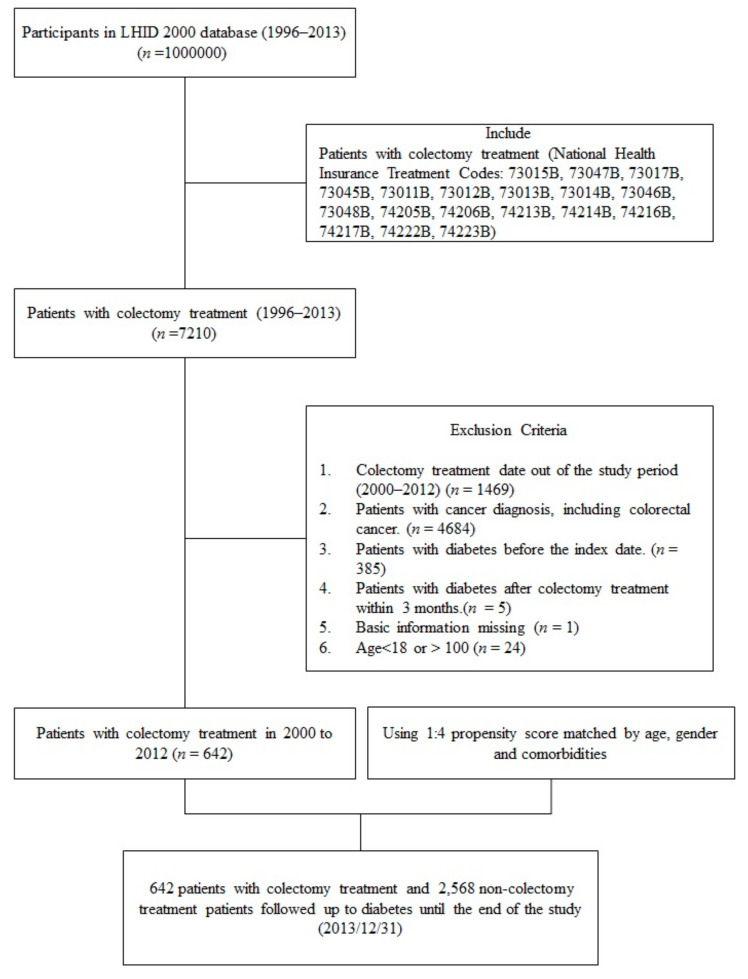
Flow chart of study cohort collection from the longitudinal database (LHID).

**Figure 2 jcm-10-05313-f002:**
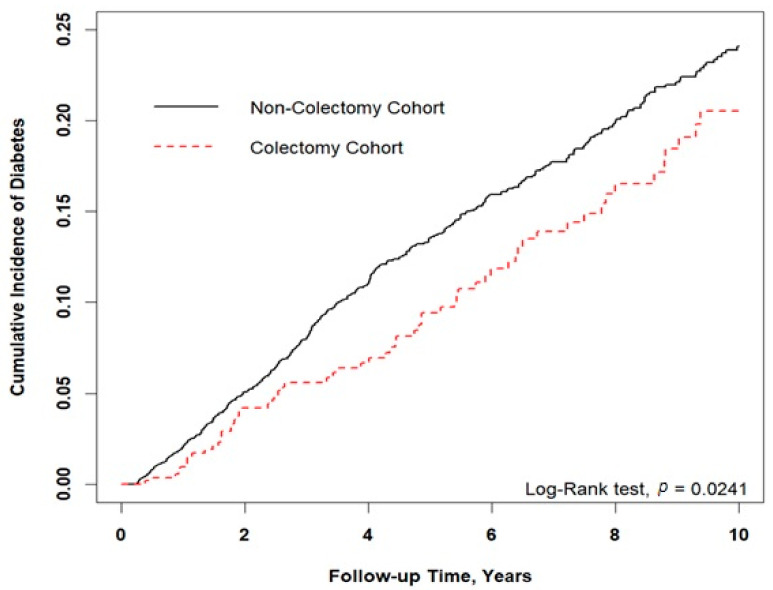
Results of Kaplan–Meier analysis for cumulative incidence of type 2 diabetes. Cumulative incidence of type 2 diabetes was lower in the colectomy cohort than the control cohort (*p* = 0.024).

**Table 1 jcm-10-05313-t001:** Demographic characteristics and comorbidities of patients who underwent colectomy in Taiwan from 2000–2012.

Variable	Total	Non-Colectomy	Colectomy	*p*-Value
	*n* = 3210	*n* = 2568	*n* = 642	
	*n*	*n* (%)/Mean ± SD	*n* (%)/Mean ± SD	
Age at baseline (years)				0.99
<40	580	463 (18.0)	117 (18.2)	
40–64	1303	1044 (40.7)	259 (40.3)	
≥65	1327	1061 (41.3)	266 (41.4)	
Mean age		58.4 (18.1)	58.3 (18.0)	0.92
Sex				0.42
Female	1219	984 (38.3)	235 (36.6)	
Male	1991	1584 (61.7)	407 (63.4)	
Baseline comorbidity				
Hypertension	1362	1095 (42.6)	267 (41.6)	0.63
Hyperlipidemia	574	459 (17.9)	115 (17.9)	0.98
Cerebrovascular disease 430–438	1253	1010 (39.3)	243 (37.9)	0.49
Heart disease	612	499 (19.4)	113 (17.6)	0.29
Thyroid disease	168	135 (5.3)	33 (5.1)	0.91
COPD	761	617 (24.0)	144 (22.4)	0.39
Renal disease	284	220 (8.6)	64 (10.0)	0.26
Liver disease	786	635 (24.7)	151 (23.5)	0.52
Anemia	462	365 (14.2)	97 (15.1)	0.56
Immune disorder	21	17 (0.7)	4 (0.6)	0.91
Follow-up time (years)		5.6 (3.6)	4.9 (4.0)	

SD: standard deviation; COPD: chronic obstructive pulmonary disease.

**Table 2 jcm-10-05313-t002:** Risk of type 2 diabetes following colectomy in patients without colorectal cancer during 14-year follow-up.

Characteristics	Event	Crude		Adjusted	
	(*n* = 65)	HR (95% CI)	*p* Value	HR ^a^ (95% CI)	*p* Value
Age at baseline					
<40	5	Ref.		Ref.	
40–64	33	3.75 (1.46–9.62)	0.006	3.39 (1.30–8.86)	0.013
≥65	27	4.56 (1.75–11.88)	0.002	4.21 (1.49–11.91)	0.007
Sex					
Female	31	1.54 (0.94–2.50)	0.086	1.72 (1.03–2.85)	0.039
Male	34	Ref.	-	Ref.	-
Baseline comorbidity					
Hypertension	30	1.96 (1.20–3.20)	0.008	1.33 (0.69–2.56)	0.391
Hyperlipidemia	15	1.68 (0.94–2.99)	0.080	1.12 (0.60–2.11)	0.723
Cerebrovascular disease	5	1.85 (1.12–3.06)	0.016	1.22 (0.63–2.37)	0.550
Heart disease	26	0.69 (0.28–1.72)	0.422	0.41 (0.16–1.09)	0.075
Thyroid disease	3	0.91 (0.28–2.88)	0.866	0.78 (0.23–2.61)	0.681
COPD	14	1.40 (0.77–2.53)	0.268	0.88 (0.46–1.68)	0.700
Renal disease	5	1.30 (0.52–3.23)	0.577	0.96 (0.37–2.49)	0.933
Liver disease	22	1.93 (1.16–3.24)	0.012	1.79 (1.02–3.13)	0.043
Anemia	9	1.32 (0.65–2.68)	0.436	1.06 (0.50–2.27)	0.872
Immune disorder	1	3.73 (0.52–26.98)	0.193	2.33 (0.27–19.91)	0.441

HR: hazard ratio; CI: confidence interval; COPD: chronic obstructive pulmonary disease. ^a^ Adjusted HR: adjusted for age and comorbidities in Cox proportional hazards regression.

**Table 3 jcm-10-05313-t003:** Incidence rates and risk of type 2 diabetes in colectomy and non-colectomy patients without colorectal cancer.

Variables	Colectomy Cases			Non-Colectomy Controls			Compared with the Non-Colectomy Controls	
	*n* = 642			*n* = 2568			Crude HR	Adjusted HR ^a^
	Event	Person Years	IR	Event	Person Years	IR	(95% CI)	(95% CI)
Overall	65	3151	20.63	398	14254	27.92	0.74 (0.57–0.96) *	0.80 (0.61–1.04)
Age at baseline (years)								
<40	5	778	6.43	30	3213	9.34	0.69 (0.27–1.77)	0.67 (0.26–1.72)
40–64	33	1396	23.63	149	6194	24.06	0.99 (0.68–1.44)	0.98 (0.67–1.44)
≥65	27	977	27.63	219	4847	45.18	0.62 (0.41–0.92) *	0.67 (0.45–1.00)
Sex								
Female	31	1172	26.45	159	5806	27.38	0.96 (0.66–1.42)	1.11 (0.75–1.63)
Male	34	1980	17.18	239	8448	28.29	0.61 (0.42–0.87) **	0.62 (0.43–0.89) **
Baseline comorbidity								
Hypertension	30	994	30.17	250	5011	49.89	0.61 (0.42–0.89) *	0.63 (0.43–0.93) *
Hyperlipidemia	15	483	31.03	90	2000	45.00	0.68 (0.39–1.17)	0.68 (0.39–1.19)
Cerebrovascular disease	5	354	14.11	93	1964	47.35	0.30 (0.12–0.73) **	0.32 (0.13–0.79) *
Heart disease	26	867	30.00	227	4647	48.85	0.62 (0.41–0.93) *	0.66 (0.44–0.99) *
Thyroid disease	3	161	18.59	11	680	16.17	1.16 (0.32–4.15)	1.11 (0.29–4.21)
COPD	14	532	26.33	116	2735	42.41	0.63 (0.36–1.09)	0.65 (0.37–1.14)
Renal disease	5	200	25.06	52	960	54.16	0.47 (0.19–1.17)	0.51 (0.20–1.31)
Liver disease	22	667	32.98	119	3428	34.71	0.97 (0.61–1.52)	1.03 (0.65–1.63)
Anemia	9	348	25.89	55	1810	30.39	0.87 (0.43–1.76)	0.89 (0.43–1.84)
Immune disorder	1	13	74.24	3	83	36.07	2.08 (0.21–20.12)	-

IR: incidence rate per 1000 person years; HR: hazard ratio; CI: confidence interval; COPD: chronic obstructive pulmonary disease. ^a^ Adjusted HR: adjusted for age and comorbidities in Cox proportional hazards regression. * *p* < 0.05; ** *p* < 0.01.

**Table 4 jcm-10-05313-t004:** Risk of type 2 diabetes following different colectomy procedures in patients without colorectal cancer.

Variable	Event/*n*	Person Years	IR	Crude HR (95% CI)	Adjusted HR ^a^ (95% CI)
Non-colectomy controls	398/2568	14254	27.92	Ref.	Ref.
Surgery procedures of the colectomy cases					
Overall	65/642	3151	20.63	0.74 (0.57–0.96) *	0.80 (0.61–1.04)
Subtotal or total colectomy	3/41	261	11.48	0.41 (0.13–1.28)	0.55 (0.18–1.72)
Right hemicolectomy, transverse colectomy	14/169	1016	13.78	0.49 (0.29–0.84) **	0.57 (0.34–0.98) *
Left hemicolectomy or sigmoid colectomy	36/294	1252	28.75	1.04 (0.74–1.46)	1.06 (0.75–1.49)

IR: incidence rate per 1000 person years; HR: hazard ratio; CI: confidence interval. ^a^ Adjusted HR: adjusted for sex, age, and comorbidities in Cox proportional hazards regression. * *p* < 0.05; ** *p* < 0.01.

## Data Availability

All personal data was removed from the data set prior to analysis according to the Taiwan Personal Data Protection Act.

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
