# Peer review of "Is Colectomy Associated with the Risk of Type 2 Diabetes in Patients without Colorectal Cancer? A Population-Based Cohort Study"

_jcm, 2021, doi:10.3390/jcm10225313_

Round 1
Reviewer 1 Report
Review of “Is Colectomy Associated with the Risk of Type 2 Diabetes in 2 Patients without Colorectal Cancer? A Population-Based Cohort 3 Study”
Chin-Chia Wu et al. describe that right or transverse colectomies may reduce the incidence of type 2 diabetes. They prudently remove the cases with colorectal cancer, because the cancer is said to be associated with type 2 diabetes. The way to recruit controls are reasonable. Although they could not add the BMI, a major confounding factor to be diabetes, the study shows very novel findings and can be a harbinger for the further study that investigate whether the colectomy can reduce the risk of diabetes without bariatric surgery. The finding is very important and basically worth to be published on the important journal for letting the results announce clinicians. Please make a modification to brush up the paper and make it greater.
Major concern:
- Please fill out the STROBE checklist for the cohort study. I think the manuscript lacks a couple of items in the checklist.
- Please describe how the study size was arrive at.
- Additionally, I would suggest to add the sentence to explain why the colorectal cancer patients should be removed like, “for the robust analysis whether the colectomy truly influence the occurrence of type 2 diabetes, the study needs to remove the effect of colorectal cancer in itself.” Please add the explanation to make more understandable.
- Please describe reasons why patients underwent colectomy in the Colectomy group.
- Reader may think if colectomy with colorectal cancer can reduce the incidence of type 2 diabetes, because colectomy is mainly performed for patients with colorectal cancer. Can you compare the incidence of type 2 diabetes between colectomy cases with and without colorectal cancer using the NHIRD database? If impossible, please describe the issue in the Discussion.
- I guess there is no description the study was approved by the IRB committee. Please describe how to cope with ethical issue for the study in the Material and Methods.
Minor concern:
- In the Introduction, the reason why colorectal cancer patients were removed from the study is a little ambiguous. For example, in line 75 “newly diagnosed type 2 diabetes was associated with a higher risk of colorectal cancer, indicating an association between colon cancer and type 2 diabetes” means colorectal cancer is relevant to incidence of diabetes or prevalence of diabetes? Entirely, the manuscript does not mention “incidence” or “prevalence”. The study is an epidemiological study and please stipulate either one in every part.
- This is a retrospective cohort study you describe in the Discussion. However, please state it in the Materials and Methods.
Reviewer 2 Report
In the present study, the authors presented “Is Colectomy Associated with the Risk of Type 2 Diabetes in 2 Patients without Colorectal Cancer? A Population-Based Cohort 3 Study”. This study is very interesting and is well analyzed. However, I think there are substantial problems about study design and discussion in this paper.
Major
- What were the causes of colectomy in 642 cases? Please analyze them. Because it may affect the development of diabetes mellitus. Also, it is probably useful for reader’s understanding.
- This is the study derived from national medical record. However, I think collected data should be examined by a physician in this kind of study whether the name of treatment and disease existed accurately.
- I think the definition of diabetes mellitus seemed vague. People who had minor diabetes mellitus without medication was not included in this definition.
- How did authors manage patients with both colectomy and gastrectomy in this study? Please explain it in the Method section.
Minor
- The sentence of a last part of Introduction is described below.
“In colonic resection for colorectal cancer, lymph node dissection offers a much more extensive eradication of metastasis than in resection for benign diseases”. I think authors would like to show the difference between colectomy for cancer and colectomy for benign disease. However, this sentence seems a little abrupt. Please change the sentence appropriately for readers understanding.
- Please amend the style of Table 4.
- There are no data about body weight after surgery.
- How did authors exclude Type 1 of diabetes mellitus?
- Authors should compare the development of diabetes mellitus of regular Taiwanese people in national statistics compared to the result of this study.
- As authors mentioned it in the limitation, BMI could not analyzed in this paper. However, BMI is associated with diabetes mellitus greatly. I ask authors to discuss more about this in the Discussion section.
Round 2
Reviewer 1 Report
The authors modified the manuscript appropriately. Only one issue is they sometimes duplicate preposition in the sentences added when revised. Please confirm their validities. Thank you for choosing me as a reviewer.
Author Response
Dear Reviewers,
Thank you for very careful review of our paper, entitled “Is Colectomy Associated with the Risk of Type 2 Diabetes in Patients without Colorectal Cancer? A Population-Based Cohort Study” with ID No. jcm-1398579 and for the comments, corrections and suggestions that ensued. We tried our best to improve the manuscript and made some changes in the paper.
Once again, thank you very much for your comments and suggestions.
Best regards,
Chun-Ming Chang
Jui-Hsiu Tsai
Reviewer 2 Report
The paper is well revised. I have no more comments.
Author Response
Dear Reviewers,
Thank you for very careful review of our paper, entitled “Is Colectomy Associated with the Risk of Type 2 Diabetes in Patients without Colorectal Cancer? A Population-Based Cohort Study” with ID No. jcm-1398579 and for the comments, corrections and suggestions that ensued.
Special thanks to you for your good comments.
Best regards,
Chun-Ming Chang
Jui-Hsiu Tsai